# EVALUATING SAE INTERPRETABILITY WITHOUT EXPLANATIONS

**Gonçalo Paulo, Nora Belrose**
EleutherAI
`{goncalo,nora}@eleuther.ai`

## ABSTRACT

Sparse autoencoders (SAEs) and transcoders have become important tools for machine learning interpretability. However, measuring how interpretable they are remains challenging, with weak consensus about which benchmarks to use. Most evaluation procedures start by producing a single-sentence explanation for each latent. These explanations are then evaluated based on how well they enable an LLM to predict the activation of a latent in new contexts. This method makes it difficult to disentangle the explanation generation and evaluation process from the actual interpretability of the latents discovered. In this work, we adapt existing methods to assess the interpretability of sparse coders, with the advantage that they do not require generating natural language explanations as an intermediate step. This enables a more direct and potentially standardized assessment of interpretability. Furthermore, we compare the scores produced by our interpretability metrics with human evaluations across similar tasks and varying setups, offering suggestions for the community on improving the evaluation of these techniques.

## 1 INTRODUCTION

Sparse autoencoders (SAEs) and transcoders are now popular tools for interpreting large language models (Lieberum et al., 2024; Gao et al., 2024; Templeton et al., 2024), vision models (Surkov et al., 2024; Gorton, 2024) and other neural networks (Adams et al., 2025; Le et al., 2024). Naively interpreting the activations of neural networks tends to fail due to polysemanticity (Mu & Andreas, 2020; Zhang & Wang, 2023) — neurons usually fire in diverse, seemingly unrelated contexts. SAEs aim to overcome polysemanticity by representing each hidden state using an overcomplete basis, while requiring the coefficients of the linear combination to be sparse and non-negative. This allows SAE basis vectors (often called *latents*) to learn more specific and interpretable latents than those captured by neurons. Initial works showed that the latents of SAEs have very interpretable activations (Cunningham et al., 2023), specially in the highest quantiles. Recent work (Karvonen et al., 2025) has focused on generating good benchmarks that go beyond minimizing the reconstruction loss and increasing sparsity, instead focusing on possible downstream applications and addressing pathologies found in some SAEs, like feature absorption (Chanin et al., 2024).

The most popular methods for evaluating SAE interpretability involves generating a natural language explanation for each latent. This explanation is then evaluated by how useful it is for predicting the activation of the latent, or the effects of intervening on the latent, in a given context (Templeton et al., 2024; Paulo et al., 2024; Gur-Arieh et al., 2025). The use of natural language explanations is problematic, however. It complicates the evaluation pipeline, introducing additional hyperparameters, such as how many examples to show the explainer model and which type of examples to show for instance. It also requires the the exploration of different prompting techniques which are likely to affect the final results, for instance allowing the model to do Chain of ThoughtWei et al. (2022), how to highlight the activating tokens as well as on which format to present the explanation - as a short sentence, a longer explanation with examples, or something in between. Ideally, we would like to abstract away from the details of explanation generation, focusing on the properties of the latents themselves.

The explanation-centered approach also assumes that, in order for a latent to be interpretable, it must have a meaning succinctly expressible in words (Ayonrinde et al., 2024; Paulo & Belrose, 2025). We

question this philosophical assumption: we deem a latent interpretable if a human can distinguish activating from non-activating examples, without producing an explanation. Given this new definition of interpretability, we introduce two methods for assessing sparse coder interpretability which do not require generating natural language explanations for latents. Our methods are applicable to both human and LLM evaluators, and we compare the scores produced by LLMs to those produced by humans

## 2 RELATED WORK

### 2.1 EVALUATING INTERPRETABILITY

We consider human evaluation to be the gold standard for interpretability, and LLM evaluators are valid only insofar as their judgments correlate strongly with those of humans. Chang et al. (2009) introduced the word intrusion task, where human participants identify an out-of-place word within a set of words associated with a specific topic discovered by a topic models. Subramanian et al. (2018) apply this approach to sparse word embeddings. The two alternative forced choice task (Borowski et al., 2020; Zimmermann et al., 2023) asks humans to discriminate between pairs of images where one strongly activates a feature map in a convolutional neural network, and one image does not. A model for which humans achieve high accuracy on these tasks, as well as a high inter-participant agreement rate, can then be considered interpretable.

Most methods of evaluating SAE interpretability involve generating an explanation for each latent, then evaluating the usefulness of these explanations for predicting whether, and/or how strongly, the latent will activate in a given context. This is similar to the simulation scoring method proposed in Bills et al. (2023) for neuron–based interpretability. However, this approach is entirely correlational, and may not capture the causal effects of latent activations on the model's output. Causal evaluations involve intervening on the model along a latent direction, and measuring how well the effect of this intervention can be predicted given a natural language explanation (Paulo et al., 2024; Gur-Arieh et al., 2025).

If SAE latents are strongly predictable by humans, we might imagine that we could replace the true latent activations from the SAE encoder with predicted activations from a human or LLM during a forward pass, and observe little performance degradation relative to the pristine model. This was attempted by Paulo & Belrose (2025), with mostly negative results. The idea is similar to concept bottleneck models (Koh et al., 2020), where the concepts are learned to explain an existing model, instead of learned as the model is trained.

### 2.2 AUTOMATIC EXPLANATION GENERATION AND EVALUATION

The standard method for generating explanations for latents involves collecting activations over a large corpus of text (Bills et al., 2023). Contexts that trigger the activation of a given latent are then presented to an LLM, which is tasked with summarizing them. Different strategies can be employed when sampling examples to show the LLM, such as stratified sampling from selected quantiles in the activation distribution (Templeton et al., 2024; Paulo et al., 2024).

Explanations can be evaluated via **simulation**, where an LLM is asked to predict a latent's activation strength at a specific token in a sentence given the explanation (Bills et al., 2023). However, this technique is computationally expensive and requires fairly heavyweight language models to produce meaningful results (Bills et al., 2023; Paulo et al., 2024). More recent methods aim to simplify this process by focusing on distinguishing between activating and non-activating examples (Paulo et al., 2024).

One such method is **detection**, where the model only needs to predict whether any token anywhere in an example activates a specified latent. In **fuzzing**, the model is asked whether a specific highlighted token in a context activates the latent or not. In Templeton et al. (2024), the authors propose a simple rubric score, where the model rates the relevance of a given interpretation in relation to the context and corresponding activations. Notably, these scoring techniques are more suitable for human evaluation due to their simplicity.

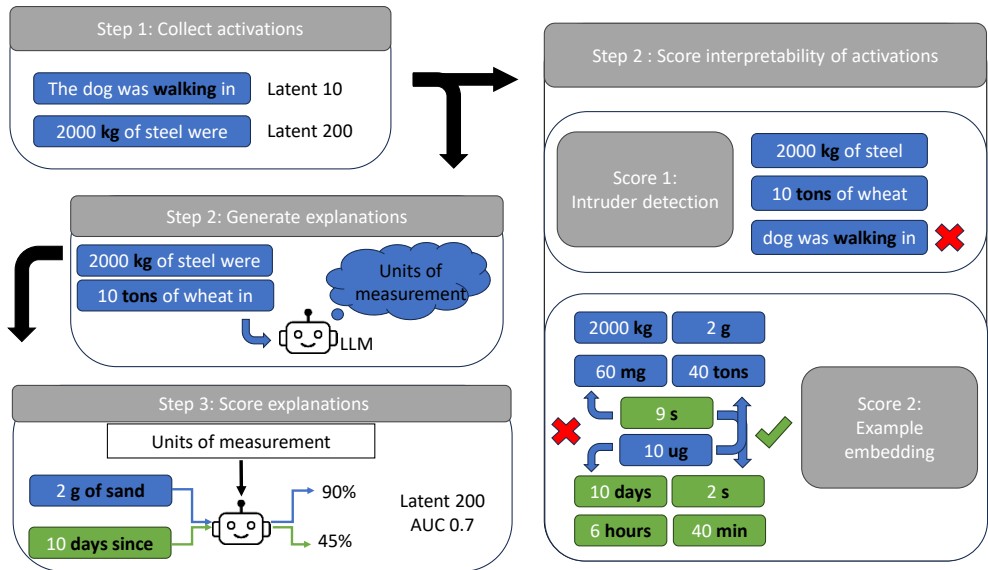

Figure 1: **Evaluating the interpretability of SAE latents.** To evaluate the interpretability of SAE latents, activations of the latents are collected over a colection of text. Traditionally these are then used to generate explanations, which are then scored. This process is an indirect measurement of the latents interpretability, and specific choices when generating explanations can influence the scores. Instead, we propose to evaluate the interpretability of SAE latents by directly looking at their activations, either through intruder detection or example embedding scoring.

Other approaches focus on the downstream effects of activating certain latents. For instance, Gur-Arieh et al. (2025) and Paulo et al. (2024) generate explanations by analyzing the impact that a latent has on the base model's output when used as a steering vector. Their scoring metrics quantify how well the explanation does at predicting these steering effects.

## 3 METHODS

### 3.1 TRAINING SAEs

In this work we focus on evaluating the interpretability of four different sparse autoencoders trained on the output of the MLPs at layers 9, 15, 21, and 27 of SmolLM2 135M (Allal et al., 2025). They were trained to minimize the mean squared error between their output and the MLP output, with no auxiliary loss terms. We adopt the state-of-the-art TopK activation function proposed by Gao et al. (2024), which directly enforces a desired sparsity level on the latent activations without the need to tune an L1 sparsity penalty, and consider the case where $k = 32$. All SAEs have 32k features. All sparse coders are trained using the Adam optimizer (Kingma & Ba, 2017), a sequence length of 2049, and a batch size of 64 sequences. We train the SAEs on 10 billion tokens sampled from a reconstruction of the SmolLM2 training corpus.

We also use our method on publicly available SAEs, namely SAEs trained on the residual stream of layer 20 of Gemma 2 9b (Team et al., 2024; Lieberum et al., 2024) (gemmascope), on the residual stream of layers 17,19 of Llama 3.1 8b (Team, 2024)(EleutherAIeleutherAI) and skip transcoders (SSTs) (Dunefsky et al., 2024) trained on the MLP of layer 6 of Pythia-160m (Biderman et al., 2023)(EleutherAI). The Gemmascope SAEs have JumpRelu activations (Rajamanoharan et al., 2024), with an average L0 of 57 and 32k features. The LLama SAEs have 131k latents and are TopK SAEs with k=32. The SSTs trained on Pythia have 32k latents and are TopK SSTs with k=64.

## 3.2 INTRUDER DETECTION

Inspired by intruder word detection (Chang et al., 2009; Klindt et al., 2025), we designed an intruder sentence detection task to evaluate the interpretability of SAE latents without relying on natural language explanations. For each latent, we sample four activating examples and one non-activating intruder example. The intruder is drawn from a pool of examples that do activate other latents but do not trigger the latent being evaluated. These examples are collect from the SAE training corpus.

To evaluate the interpretability of each latent, we present the five examples to the LLM as a numbered list, and ask it to identify the intruder by index. Activating tokens in the activating examples are highlighted using $<<$ and $>>$ to provide additional context, while in non-activating examples, a random selection of tokens is highlighted. The number of highlighted tokens matches the average count in the activating examples, rounded down. Each example is constrained to 32 tokens in length to maintain consistency. Examples are preceded by a few-shot prompt demonstrating the task on synthetic examples.

The activating examples show are chosen at random from one of ten deciles of the activation distribution. Decile sampling is done for two reasons: first there has been previous work (Paulo et al., 2024) that showed that using decile sampling one could evaluate the interpretability of different parts of the distribution. We are also interested in understanding the similarity of the activations of the same feature, and dividing the distribution into bins makes it easier for us to compare different parts of the activation distribution.

The score of the latent is then defined as the accuracy of the evaluator at intruder detection for that latent, averaged over deciles. We also report the accuracy for individual deciles, allowing for the evaluation of interpretability as a function of activation strength.

We depart from traditional intruder word detection by showing a full context, because we find that a significant number of latents activate on more than one token in a context, sometimes on several adjacent tokens.

### 3.2.1 HUMAN INTRUDER DETECTION

The authors manually performed the intruder task on a subset of examples to provide a human baseline. The procedure is similar to what is done by the LLM. The authors were shown five examples, with tokens highlighted as described above, and had to select which of the examples is the intruder. In LLMs, each intruder test is done in a fresh context, and so there is no information that carries from one set of examples to the next. The same is not true for the human participants. To minimize this effect, we randomly select which latent to sample the examples from, such that there aren't many consecutive examples of the same latent. The decile from which the examples are drawn are also chosen randomly and not shown to the participants.

### 3.3 EXAMPLE EMBEDDING SCORING

Although intruder detection is able to skip the explanation generation process, we find that a powerful LLM is still needed to produce meaningful scores. Since high-quality embedding models can be quite small, we explore a method we call example embedding scoring to speed up the evaluation process. It is inspired by the two alternative forced choice task used in Borowski et al. (2020); Zimmermann et al. (2024).

Example embedding scoring evaluates the interpretability of a latent by measuring whether activating examples cluster in embedding space. We sample sentence embeddings for a set of activating examples $E^+ = \{\mathbf{e}_1^+, \dots \mathbf{e}_n^+\}$ and a set of non-activating examples $E^- = \{\mathbf{e}_1^-, \dots \mathbf{e}_n^-\}$. We also sample embeddings for an activating example $\mathbf{q}_+$ called the **positive query**, and a non-activating example $\mathbf{q}_-$ called the **negative query**. We compute how much closer each query is to examples

from its own class than to examples from the other class:

$$\Delta_+ = \frac{1}{N}\Big(\sum_{\mathbf{e}_i^+ \in E^+} \frac{\mathbf{q}^+ \cdot \mathbf{e}_i^+}{\|\mathbf{q}^+\|\|\mathbf{e}_i^+\|} - \sum_{\mathbf{e}_i^- \in E^-} \frac{\mathbf{q}^+ \cdot \mathbf{e}_i^-}{\|\mathbf{q}^+\|\|\mathbf{e}_i^-\|}\Big) \tag{1}$$

$$\Delta_- = \frac{1}{N}\Big(\sum_{\mathbf{e}_i^- \in E^-} \frac{\mathbf{q}^- \cdot \mathbf{e}_i^-}{\|\mathbf{q}^-\|\|\mathbf{e}_i^-\|} - \sum_{\mathbf{e}_i^+ \in E^+} \frac{\mathbf{q}^- \cdot \mathbf{e}_i^+}{\|\mathbf{q}^-\|\|\mathbf{e}_i^+\|}\Big) \tag{2}$$

As in intruder detection, all positive examples are sampled from the same decile of the activation distribution, and the negative examples are random non-activating contexts. All examples are sourced from the examples found in the training data of the SAE.

$\Delta_+$ and $\Delta_-$ can used to compute the AUROC, which is the scoring metric for this method. For each latent we iterate over different sets of explanations and queries, and average over them. This metric is similar to that of embedding scoring (Paulo et al., 2024), where explanations are substituted by activating examples.

### 3.3.1 FINETUNING AN EMBEDDING MODEL

We found that off-the-shelf embedding models did not perform the example embedding task very well, and we suspect that part of it is that they can't infer the meaning of the highlighting tokens we use to indicate which tokens are active in a sentence, see Table A1. We chose to finetune the stella_en_400M_v5 embedding model, as it was lightweight but demonstrated good performance in embedding benchmarks. We selected latents from the TopK skip transcoders trained on the MLP of layer 6 of Pythia-160m, with k=32(, as these would be not be used in this study and so would reduce the chances that the model would be overfit to contexts it had seen. First, we filter latents that have a score higher than 0.7 on both fuzzing and detection, until we have c.a 300 features. Secondly we create contexts similar to the ones used in example embedding, choosing 9 activating examples and an activating query and 9 non activating examples and a non activating query, with 100 of these sets for each feature. We finetune using Multiple Negatives Ranked Loss, which takes pairs of (query, positive example) and increases their similarity with respect with all the other elements in the batch. This is not exactly the same task as example embedding but we find that it approximates it in a reasonable manner.

## 4 RESULTS

### 4.1 INTRUDER DETECTION

The average intruder detection accuracy for a human is 64%, as measured on 105 different latents. On the highest activating deciles, human accuracy averaged 78%. In total, one third of latents had a score of over 80%, and only one in seven had a score lower than 30%, meaning that they few were considered non-interpretable. This supports the claim that explicit natural language interpretations are not needed to decide wether a given example should be active or not, and that showing other activating examples is sufficient.

To validate the usage of LLMs instead of humans in the evaluation process, we compute the scores given by the LLM on the same latents. While humans only see on average 10 to 20 prompts per latent, Claude Sonnet 3.5, the LLM we compare to in the left panel of Figure 2, is evaluated on 50 prompts. The Spearman correlation between the LLM scores and the human score is 0.83, a higher agreement than what was seen in previous SAE evaluation metrics (Paulo et al., 2024), although in those works the human evaluators were not performing exactly the same task, as is the case here. The expected accuracy of random guessing is 20%, so each latent that has intruder detection accuracy of 20% or less is probably uninterpretable just by looking at activating contexts. We create four additional interpretability bins, ranging from a low degree of interpretability (20-40%) to a very high degree of interpretability (80-100%). In the right panel of Figure 2 we report the agreement between binned human and binned LLM scores. We observe that the human mostly finds latents more interpretable than their LLM counterparts. This is good news, as it suggests LLMs are not finding convoluted patterns in the latent activations that humans cannot find.

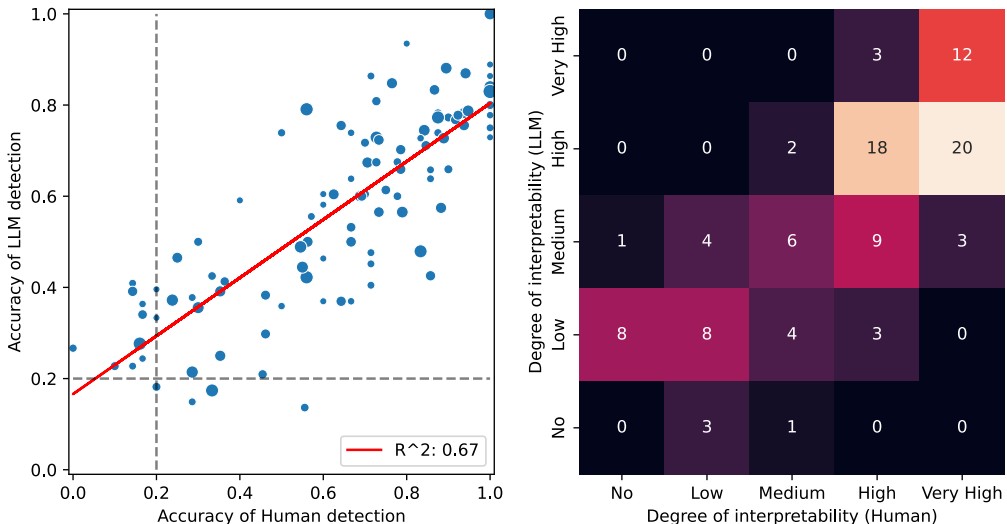

Figure 2: **Correlation between human and LLM intruder detection accuracy**. In the left panel we compare the accuracy on the intruder task for the LLM - in this case Claude Sonnet 3.5 - and that of a human. In the right panel we show a more coarse grained classification. All latents which have less than that 0.2 accuracy are considered non interpretable, and different degrees of interpretability are assigned to the other 4 bins of 0.2. The agreement between the "quality" of the latents given by humans and the LLM is significant, even though the LLMs underestimate the degree of interpretability. We evaluate 105 latents from SAEs trained on SmolLM 2 - around 26 latents per layer on four different layers. The size of each dot represents how many prompts the human saw for each latent, and is meant to represent the uncertainty associated with the score, with the smaller dots representing 8 prompts.

Accuracy on the intruder task is dependent on the the activation decile from which the examples were sampled. Examples from the highest activating decile can have up to 20% higher accuracy than examples from the smallest activations (Figure 3 left panel).

Previous work (Templeton et al., 2024; Paulo et al., 2024) has found that examples with the lowest activations are hardest to interpret, and we observe the same thing. On the other hand, even though intruder detection accuracy is lower in the lower activating deciles, is still significantly higher than random. If we divide latents into interpretability bins, we see that the most interpretable latents remain interpretable even in the lower activation deciles, with accuracies greater than 0.75 (Figure A1), which we believe show that even low activating examples can carry information about the behaviour of the latent.

### 4.1.1 DETECTING INTRUDER DECILES

Intruder detection makes it easy to investigate how the *meaning* of a latent varies as a function of the activation strength. Specifically, we modify our setup so that all five examples contain a token that activates the same latent, but one of them, the intruder, is sampled from a different activation decile as the rest. For a perfectly monosemantic latent representing a **binary** feature, we would expect the accuracy of this task to be close to random, as examples from different deciles would be very similar. If we assume that some latents represent **scalar** features with a notion of degree or intensity, one would expect that nearby deciles would exhibit close to random accuracy, and that distant pairs of deciles would be easier to distinguish. We would also expect symmetry, where the difficulty of finding a high-activating intruder among low-activating examples is similar to the accuracy of finding a low-activating intruder among high-activating examples.

In the right panel of Figure 3, we use Llama 3.1 70b (Team, 2024) to perform this detection task. As expected, Llama achieves highest accuracy when the majority decile is highest and the intruder decile is lowest (51%). This result is far from symmetric, however: when the majority decile is low,

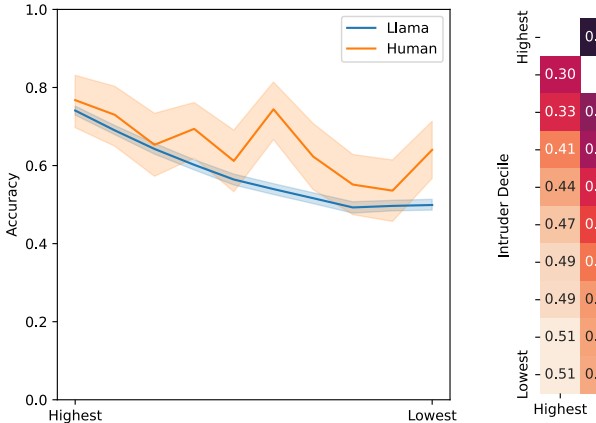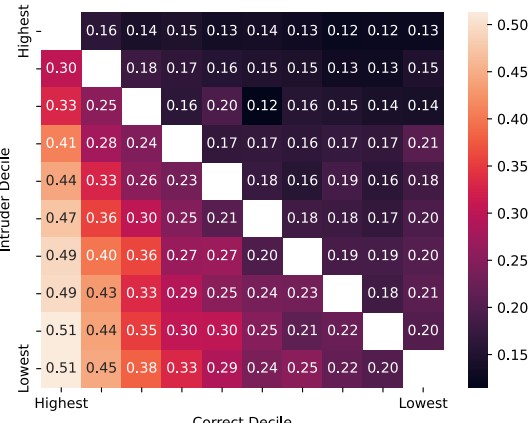

Figure 3: **Interpretability of the distribution of activations.** (Left) The activation examples used on the intruder task come from different deciles of the distribution of the latent's activations. On the lower activating deciles, it's harder to distinguish which is the intruder example, but the accuracy is still significantly above random. This shows that the full distribution of the activations of the SAE latents remains somewhat interpretable. We evaluate 105 latents using human labeling, and 1800 latents using the LLM judge (Llama 3.1 70b). (right) If instead of using non-activating examples we use examples from a different decile, we can observe that the accuracy of close deciles is close to random, while it increases for further deciles, remaining lower than when using non activating examples. We evaluate 3700 latents using the LLM judge (Llama 3.1 70b).

Llama essentially never achieves higher than random accuracy, no matter the decile of the intruder. Interestingly, when the intruder decile is high and the majority decile is low, we see significantly worse than random accuracy (around 13%).

While there likely are some 'perfect' scalar features in these SAEs, we do not see clear evidence for them in the right panel of Figure 3. The accuracy matrix is nonsymmetric, and even in the lower triangular it is not completely explained by the distance between majority and intruder deciles except by the highest activating deciles. We can rule out the hypothesis that most features are both binary and monosemantic, since we see far better than random accuracy for many (intruder, majority) pairs.

### 4.1.2 INTER-LLM AND INTER-HUMAN AGREEMENT

We investigate how the intruder accuracy for a given latent is correlated across different LLMs. We find that different models' scores have around the same correlation than with humans, except for the smaller Llama 3.1 8b, which has the worst accuracy (27%), and the lowest correlation with human judgment.

We are also interested in inter-human agreement. We find that, in a smaller set of 40 latents, each with only 3-5 prompts, the correlation between the two human labelers is of 0.87, while the one between Claude-Sonnet and each of the humans is 0.86 and 0.69. This level of inter-LLM and inter-human agreement is promising for this proposed method of evaluating SAE latent interpretability. In Table 4.1.2 we observe that scores given by humans, computed on 105 latents, have high correlation scores, >0.80, for several strong LLMs, but that it is lower for a weaker LLM.

### 4.2 EXAMPLE EMBEDDING SCORING

Example embedding scoring also finds that high deciles are easier than from low deciles (Figure 4 left panel). Unlike intruder detection accuracy, example embedding scoring is symmetric by construction (Figure 4 right panel). Example embedding scoring also finds that highly activating deciles are easy to tell apart from random examples, but has an harder time to distinguish between examples

|  | Human | Llama 3.1 70b | Llama 3.1 8b | QwQ 32b | Gemini Flash 2.0 | Claude Sonnet 3.5 |
|---|---|---|---|---|---|---|
| Human | 1 | 0.76 | 0.52 | 0.78 | 0.80 | **0.83** |
| Llama 3.1 70b | 0.76 | 1 | 0.58 | **0.89** | 0.85 | 0.88 |
| Llama 3.1 8b | 0.52 | 0.54 | 1 | 0.59 | **0.60** | 0.57 |
| QwQ 32b | 0.78 | 0.80 | 0.59 | 1 | 0.86 | **0.90** |
| Gemini Flash 2.0 | 0.80 | 0.85 | 0.60 | 0.86 | 1 | **0.87** |
| Claude Sonnet 3.5 | 0.84 | 0.88 | 0.57 | **0.90** | 0.87 | 1 |

Table 1: **Correlation between different evaluators.** We measure the Pearson correlation between the accuracy of intruder detection in 56 latents. Each latent was evaluated in 10-20 prompts by the human, and on 100 prompts by the LLMs.

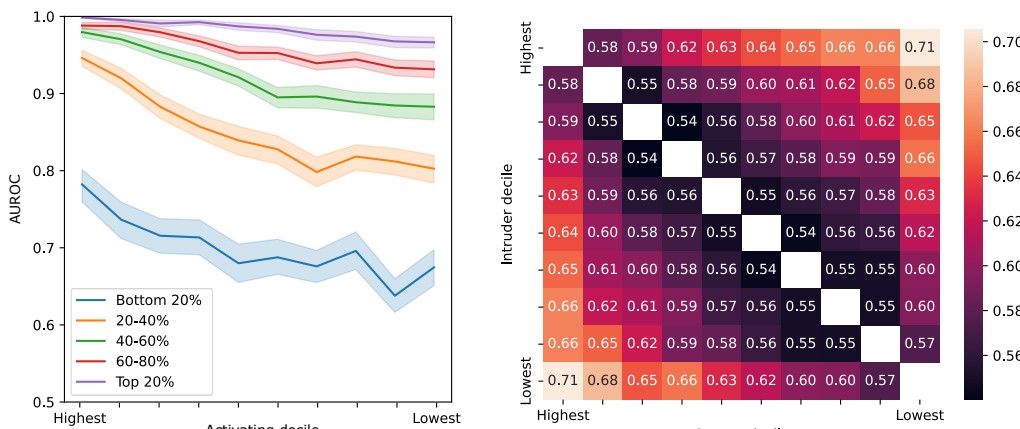

Figure 4: **Interpretability evaluation using example embedding scoring.** In the left panel, we show that the highest activating decile are easier to distinguish from non activating examples, as was the case with intruder detection. On the panel on the right, it is possible to see that, if using examples from different deciles, close deciles are hard to distinguish, while further deciles the accuracy slightly increases above random, up to 0.7 AUC for some distances. We evaluate this method on 700 latents.

belonging to the same feature but from different deciles, which the AUC for the largest possible difference reaching only 0.7.

Example embedding scores correlate as strongly with human intruder scores ($r = 0.78$) as LLM intruder scores do ($r = 0.75$). See Table 2 for full results. Due to it being fast and computationally efficient technique relative to other approaches, it might be an appropriate technique for large scale evaluation of SAEs.

In Tables A2 and A3 we show that SAEs trained in different models, with different activation functions, training objectives and sparsity levels can be evaluated using our methodology, by showing they have similar levels of correlations to what is shown in Table 2.

## 5   CONCLUSION

Prior work in the evaluation of SAEs relied on generating natural language explanations for SAE latents, which were then used to predict either the activations of those latents, or their causal effects. In this work we introduced two new methods for evaluating SAEs which do not rely on explanations: intruder detection, and example embedding scoring. We tested intruder detection using both human and LLM evaluators, finding that the human accuracies are highly correlated with those from LLMs. We use both intruder detection and example embedding scoring to measure the interpretability of different parts of the activation distribution, finding that higher deciles are more interpretable. As

| | Example Embedding | Intruder Detection (Human) | Intruder Detection (LLM) | Fuzzing (LLM) | Detection (LLM) |
|---|---|---|---|---|---|
| **Example Embedding** | 1.0 | 0.78 | 0.77 | **0.87** | 0.52 |
| **Intruder Detection (Human)** | **0.78** | 1.0 | 0.75 | 0.77 | 0.48 |
| **Intruder Detection (LLM)** | 0.77 | 0.75 | 1.0 | **0.78** | 0.56 |
| **Fuzzing (LLM)** | **0.87** | 0.77 | 0.78 | 1.0 | 0.64 |
| **Detection (LLM)** | 0.52 | 0.48 | 0.56 | **0.64** | 1.0 |

Table 2: **Correlation between different methods.** We measure the Pearson correlation between the accuracy of intruder detection and other scoring methods in 105 latents. Each latent was evaluated in 10-20 prompts by the human, and on 100 prompts by the other methods.

expected, example embedding scoring has a lower correlation with human intruder detection scores than LLM intruder detection scores do, but we believe it to be a promising method due to its speed: it uses a small embedding model rather than a large language model to perform the scoring.

## 6 ACKNOWLEDGMENTS

We would like to acknowledge our discussion with David Klindt that initiated this work. We are thankful to Open Philanthropy for funding this work. We are grateful to CoreWeave for providing the compute resources.

## AUTHOR CONTRIBUTIONS

Gonçalo Paulo developed the scoring methods and performed the experiments. GP wrote the initial draft. Nora Belrose performed some intruder detection experiments, provided guidance and did major reviews to the draft.

## CODE AVAILABILITY

Both the intruder detection scorer and the example embedding scorer can be used in our pipeline for automated interpretability of sparse coders, delphi. Sparse autoencoders where trained using our training library, sparsify

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

# A APPENDIX

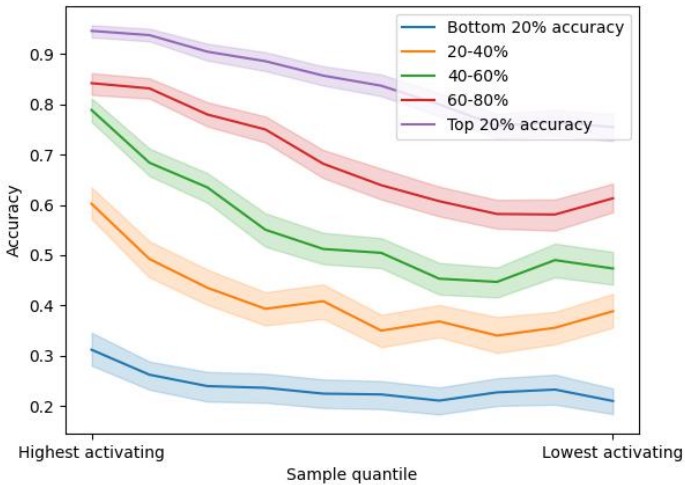

Figure A1: **Interpretability of the distribution of activations, stratified.** If we observe each of the 5 bins of accuracy separately, the bin with higher accuracy consistently has high accuracy throughout all the distribution, while the other bins have sharper drops in accuracy when going from the highest activating examples to the lowest ones.

| | Example Embedding | Example Embedding (no finetuning) | Intruder Detection (Human) | Intruder Detection (LLM) | Fuzzing (LLM) | Detection (LLM) |
|---|---|---|---|---|---|---|
| **Example Embedding** | 1.0 | 0.47 | 0.78 | 0.77 | 0.87 | 0.52 |
| **Example Embedding (no finetuning)** | 0.47 | 1.0 | 0.42 | 0.46 | 0.50 | 0.60 |
| **Intruder Detection (Human)** | 0.78 | 0.42 | 1.0 | 0.75 | 0.77 | 0.48 |
| **Intruder Detection (LLM)** | 0.77 | 0.46 | 0.75 | 1.0 | 0.78 | 0.56 |
| **Fuzzing (LLM)** | 0.87 | 0.50 | 0.77 | 0.78 | 1.0 | 0.64 |
| **Detection (LLM)** | 0.52 | 0.60 | 0.48 | 0.56 | 0.64 | 1.0 |

Table A1: **Correlation between different methods.** We measure the Pearson correlation between the accuracy of intruder detection and other scoring methods in 112 latents. Each latent was evaluated in 10-20 prompts by the human, and on 100 prompts by the other methods.

|  | Example Embedding | Intruder Detection (LLM) | Fuzzing (LLM) | Detection (LLM) |
|---|---|---|---|---|
| **Example Embedding** | 1.00 | 0.84 | 0.90 | 0.58 |
| **Intruder Detection (LLM)** | 0.84 | 1.00 | 0.83 | 0.67 |
| **Fuzzing (LLM)** | 0.90 | 0.83 | 1.00 | 0.71 |
| **Detection (LLM)** | 0.58 | 0.67 | 0.71 | 1.00 |

Table A2: **Correlation between different methods in Gemmascope SAEs trained onGemma 2 9b.**We measure the Pearson correlation between the accuracy of intruder detection and other scoring methods in 50 latents. Each latent was evaluated on by 100 prompts for each method.

|  | Example Embedding | Intruder Detection (LLM) | Fuzzing (LLM) | Detection (LLM) |
|---|---|---|---|---|
| **Example Embedding** | 1.00 | 0.67 | 0.84 | 0.40 |
| **Intruder Detection (LLM)** | 0.67 | 1.00 | 0.70 | 0.55 |
| **Fuzzing (LLM)** | 0.84 | 0.70 | 1.00 | 0.59 |
| **Detection (LLM)** | 0.40 | 0.55 | 0.59 | 1.00 |

Table A3: **Correlation between different methods in SSTs trained on Pythia-160m.**We measure the Pearson correlation between the accuracy of intruder detection and other scoring methods in 230 latents. Each latent was evaluated on by 100 prompts for each method.

