# OpenReview forum: "Evaluating SAE interpretability without generating explanations"
_ICLR.cc/2026/Conference — ICLR 2026 Poster_

### Official Review · Reviewer_KYqC · 2025-10-30

**Soundness:** 4
**Presentation:** 2
**Contribution:** 3
**Rating:** 4
**Confidence:** 4

**Summary:**

The paper introduces a new method for evaluating Sparse Autoencoders (SAEs). It argues that explaining latent directions in the SAE’s latent space through short textual descriptions is suboptimal for two main reasons. First, this approach complicates the evaluation process by adding hyperparameters and prompt-related variability. Second, a latent factor can be interpretable even if it cannot be concisely expressed in words.

As an alternative, the paper proposes an intruder detection framework. For each latent, four activating examples and one non-activating “intruder” example are sampled. Interpretability is then assessed based on how effectively humans, large language models (LLMs), and an embedding-based algorithm can identify the intruder. This approach emphasizes intuitive recognition of the pattern a latent does or does not encode.

The results show strong agreement between human and LLM performance in intruder detection, indicating that LLMs may be well-suited for automating SAE interpretability evaluation.

**Strengths:**

Proposes new method for evaluating SAEs that is more permissive in the types of interpretability it allows for (interpretable, but not easily expressible in words).

The proposed method looks very promising, with LLM accuracies tracking those of humans.

Multiple approaches toward the task are evaluated (LLM vs. embedding).

**Weaknesses:**

I think the presentation could be significantly improved.
On line 155, it is explained that 'We randomly select one of the ten deciles of the activation distribution, then sample all of our activating examples from the same decile.', but this is then not at all motivated. I found it quite difficult to understand why we would want to do this, and it wasn't until re-reading some of the results section for the second time that I understand the point. Specifically, the paragraph on lines 295-304 goes into the different ways we might (not) assign meaning to the activation strengths of the latent. I think that goes a long way towards motivating why we care about deciles, but it appears in the results section, rather than in an earlier section, where I would expect it.

**Questions:**

Looking at the interpretability of distributions of activations, the LLM's results are very far from symmetric:
it is much better at detecting a low-activating intruder among highly-activating samples than vice versa.
This is something I could not have predicted, do you have any intuition for why this is? And, do you have any data on how symmetrical humans are?

Have you compared your interpretability scores to the explanation-centered approach you contrast to in the introduction? Can you find examples of latents which would be deemed uninterpretable according to other methods, but are considered interpretable under your framework?

---

> ### Author Response · Authors · 2025-11-25
>
> > ‘On line 155, it is explained that 'We randomly select one of the ten deciles of the activation distribution, then sample all of our activating examples from the same decile.', but this is then not at all motivated.’
>
> We have now expanded that section, motivating this choice in two ways. First, we wanted to be able to measure the interpretability of different parts of the distribution, and we used the technique introduced in ‘Automatically interpreting Millions of Features’ where the distribution of activations is divided into equally sized bins. Secondly, we wanted to measure how similar the quantiles were to one another (the experiments described in 4.1.1).
>
> > Looking at the interpretability of distributions of activations, the LLM's results are very far from symmetric: it is much better at detecting a low-activating intruder among highly-activating samples than vice versa. This is something I could not have predicted, do you have any intuition for why this is? And, do you have any data on how symmetrical humans are?
>
> Intuitively, high-activating examples are all similar to one another because they are all close to the most prototypical example of the feature. By contrast, low-activating examples are all far from the prototype, and they differ from the prototype in idiosyncratic ways depending on the specific example. So low-activating examples tend to be far from one another. When the intruder is a low-activating example, it is easy to distinguish it from the rest because it is semantically distant from them, while the rest are close to one another. By contrast, when the intruder is a high-activating example, it does not stand out from the rest, because all of the examples are roughly equally dissimilar to one another. We don’t have data on how symmetrical humans are. To have enough samples to have results as strong as the ones we produce with LLM judges would require more than 10x our human labeling.
>
> > Have you compared your interpretability scores to the explanation-centered approach you contrast to in the introduction? Can you find examples of latents which would be deemed uninterpretable according to other methods, but are considered interpretable under your framework?
>
> We find some examples that have low Fuzzing score but high Embedding score, but not the reverse. We also find examples that have a high Fuzzing score and a low Intruder score. We would be happy to add such examples to the appendix if the reviewer believes this would improve the paper.

---

> > ### Comment · Reviewer_KYqC · 2025-11-27
> >
> > Thank you for your response. I think the changes address my concerns, and your explanation for question 2 makes a lot of sense. I have raised my score correspondingly. I believe adding the examples to the appendix would be great, although if there is space, I personally think they would not be out of place in the paper itself.

---

### Official Review · Reviewer_6pUU · 2025-10-30

**Soundness:** 3
**Presentation:** 2
**Contribution:** 2
**Rating:** 4
**Confidence:** 3

**Summary:**

This paper proposes a novel evaluation approach to assess the interpretability of sparse autoencoders (SAEs). Instead of generating natural language explanations as an intermediate step, the authors introduce two explanation-free methods: intruder detection and example embedding scoring. The paper demonstrates that direct assessment of latent interpretability is viable and correlates well with human judgments when using an LLM-as-a-judge approach.

**Strengths:**

- This paper demonstrates the feasibility of the proposed method intruder detection, achieving strong correlation between human and LLM assessments.
- The methods used in this paper are straightforward and easy to understand.
- The paper examines interpretability across different activation deciles, providing nuanced insights into how interpretability varies with activation strength.

**Weaknesses:**

- However, the performance of embedding score is not promising. The AUROC scores are barely above random (0.5-0.7), and correlation with human judgments is weak (r=0.48). This undermines one of the paper's main contributions, as this method was proposed as a fast, scalable alternative.
- Lack of direct performance comparison with traditional interpretability evaluation methods.
- Results are presented on very small set of latents (56) and small models. So we don't know if this holds when dataset scales up.
- The bottleneck of evaluation seems to be extensive data collecting process, why avoiding natural language explanation is a critical problem?
- Despite claiming to simplify evaluation, intruder detection still relies heavily on LLM queries, contradicting the motivation of reducing computational costs.

**Questions:**

- Line 34, the coefficients is not non-negative necessarily, if this refers to activation value of latents. Please verify with examples from Neuronpedia.
- Line 44 - 46, the conclusion on natural language explanations introduced additional hyper parameters and prompts can be expanded further. It’s not very clear how they introduce additional parameters, which might refer to simulations. But it’s important to explain this clearly at the beginning of the paper. I feel the authors should spend more time polishing the introduction section to stand out their motivation and make it accessible. The last paragraph of the introduction is hard to follow. The introduction to their own methods is not clear at all.
- Heatmap in Figure 2 is not very illustrative. What does "All latents which have less than that 0.2 accuracy are considered non interpretable, and different degrees of interpretability are assigned to the other 4 bins of 0.2." mean? Would overlapping histogram be more illustrative here?

---

> ### Author Response · Authors · 2025-11-25
>
> We would like to thank the reviewer for their questions and time spent to give us feedback. We will start to address the different weaknesses raised and then answer the questions.
>
> > ‘However, the performance of embedding score is not promising.  (...)’
>
> We apologize for the confusion about the AUROC scores. The scores are only close to random in the special case where examples that activate the same feature, but in a different activation decile, are used as intruders. As we mentioned in other reviews, finetuning the embedding model on the task greatly increased the correlation with human scores.
>
> > Lack of direct performance comparison with traditional interpretability evaluation methods.
>
> Could the reviewer clarify what is meant with performance on this question?
>
> > Results are presented on very small set of latents (56) and small models. So we don't know if this holds when dataset scales up.
>
> We again apologize for the misunderstanding. Only the human trials were using evaluating only 56 latents. We have expanded that number to the best of our current ability to 105. The other experiments are all based on at least 100 latents per SAE evaluated. We have also expanded our work to other SAEs, specifically of larger models like Gemma 9b and Llama 8b.
>
> > The bottleneck of evaluation seems to be extensive data collecting process, why avoiding natural language explanation is a critical problem?
>
> Collecting activations on 10M tokens on most SAEs only takes a couple of minutes for the small models, some hours for the larger models - this for all the latents. It also only requires forward passes on the model that we are investigating. On the other hand generating an explanation, while taking only a few seconds per latent, ends up requiring several days if one wants to explain all features, and the querying of larger models than the ones being investigated with the SAEs. Not only that, but even if explanation generation was not a real computational bottleneck it makes evaluation less reliable, because it is hard to distinguish if problems are on the explanation generation pipeline or if the SAEs is hard to interpret by itself.
>
>
> > Line 34, the coefficients is not non-negative necessarily, if this refers to activation value of latents. Please verify with examples from Neuronpedia.
>
> We are unsure what the reviewer means here. TopK SAEs use ReLU before the TopK operation, so all coefficients are non-negative. Most recent architectures also use some kind of ReLU in their pre-activations. Which examples in Neuronpedia have negative activations?
>
> > Line 44 - 46, the conclusion on natural language explanations introduced additional hyper parameters and prompts can be expanded further. It’s not very clear how they introduce additional parameters, which might refer to simulations. But it’s important to explain this clearly at the beginning of the paper. I feel the authors should spend more time polishing the introduction section to stand out their motivation and make it accessible.
>
> We have added some examples of hyperparameters and prompt choices that are required when generating an explanation.
>
>
> > Heatmap in Figure 2 is not very illustrative. What does "All latents which have less than that 0.2 accuracy are considered non interpretable, and different degrees of interpretability are assigned to the other 4 bins of 0.2." mean? Would overlapping histogram be more illustrative here?
>
> Given that there are 5 examples in any intruder detection sample, having an accuracy of c.a 0.2 is equivalent to guessing. Because of this we consider latents uninterpretable. Inspired by the specificity rubric of ‘Scaling Monosemanticity: Extracting Interpretable Features from Claude 3 Sonnet,’ we bin the interpretability scores of our latents into 5 bins. Latents that have accuracies between 0 and 0.2 are not interpretable, latents that have accuracies between 0.2 and 0.4 have a low degree of interpretability, those between 0.4 and 0.6 have a medium degree, and so on.

---

> > ### Comment · Reviewer_6pUU · 2025-11-27
> >
> > I thank the authors for their detailed rebuttal and the significant effort to expand the human evaluation to 105 latents. I apologize for my factual error regarding coefficient values; I conflated the logits visualized on Neuronpedia with the actual activations, and I accept the clarification regarding the embedding score's validity. The authors have convincingly addressed my concerns regarding sample size robustness and the computational motivation for their metric.
> >
> > Therefore, I am raising my score.

---

### Official Review · Reviewer_Bo6j · 2025-11-01

**Soundness:** 3
**Presentation:** 4
**Contribution:** 3
**Rating:** 4
**Confidence:** 5

**Summary:**

This paper proposes two explanation-free methods for evaluating the interpretability of sparse autoencoders (SAEs): intruder detection and example embedding scoring. The authors test the proposed methods on SmolLM2 135M across 56 latents and find a strong correlation between human and LLM evaluators in intruder detection. The intruder detection method successfully bypasses natural language explanation generation while maintaining interpretability assessment; however, the embedding method shows limited correlation with human judgments. Higher activation deciles prove more interpretable across both methods, and the evaluation reveals that most SAE latents demonstrate interpretability without requiring explicit verbal descriptions.

**Strengths:**

1. Figure 1 effectively illustrates the conceptual shift from explanation-based to activation-based evaluation, and the writing is generally accessible.

2. The paper introduces evaluation methods that bypass natural language explanation generation, addressing a significant limitation in existing sparse autoencoders' interpretability assessment. This is a significant contribution that streamlines the evaluation pipeline and minimizes the impact of confounding factors.

3. Example embedding scoring offers a computationally lightweight alternative using small embedding models, making large-scale SAE evaluation more feasible.

**Weaknesses:**

1. The evaluation focuses exclusively on SmolLM2 135M across only 4 layers with 56 total latents. This narrow scope raises questions about generalizability to larger models, different architectures, or other SAE training approaches beyond TopK.

2. Example embedding scores do not correlate as strongly with human intruder scores (r = 0.48), and AUROC are close to random, which limits the practical utility of the proposed scoring method.

3. The paper lacks a discussion of failure modes or which types of latents are poorly captured by the proposed methods.

4. Limited investigation of why LLMs consistently underestimate interpretability compared to humans

**Questions:**

1. Why does the example embedding score not correlate as strongly with human intruder scores as it does with LLM intruder scores? Authors say example embedding scores tend to underestimate the interpretability of latents due to the small size of the embedding. Does the correlation improve when the embedding size is increased?


2. How sensitive are the intruder detection results to the highlighting strategy? Have you tested alternative approaches, such as not highlighting any tokens or using attention-based highlighting to focus on the most relevant tokens?

3. What is the rationale for randomly selecting a single decile and sampling all activating examples from it?

4.  Proposed SAEs use TopK activation with $k=32$. How do results change with different $k$ values, different activation functions, or different sparsity levels?

5. Can you provide examples of latents that score poorly on intruder detection but might still be considered interpretable by other measures?

---

> ### Author Response · Authors · 2025-11-25
>
> We would like to thank the reviewer for the time spent to provide a thorough review. We will reply to the different weaknesses and questions by their respective numbers.
>
> On **weakness 1)** we apologize for not making it clear that for most of our results, we actually were using at least 100 latents for each of the 4 layers for most experiments, with more than 1000 total latents in some of them. We have updated the text to clarify the number used. We also have increased the number of human labeled latents to 100 and have expanded our analysis to include latents from SAEs trained in Gemma and Pythia. The Gemma SAEs are from Gemmascope, which uses JumpReLU as the activation function. Due to time constraints we are not able to produce human correlations for all those models, but we can provide correlations between those scores and other established scores (fuzzing and detection). We observe that for these other models and architectures, the correlations are very similar to those on SmolLM2.
>
> On **weakness 2)** and **question 1)** we want to highlight that the close-to-random AUC is only for the case where the intruder comes from the same activating feature, but from a different quantile. With the updated text and results we think this is no longer a confusing part about our test. We have also found that the correlation of the embedding score with the other score can be improved by fine tuning the embedding model to better perform this task on a set of held out features.
>
> **Weakness 3) Question 5)**: We find some examples that have low Fuzzing score but high Example embedding score (but not the opposite) as well as examples that have a high Fuzzing score and a low intruder detection score. Would the reviewer like to see some example dashboard, for instance similar to that of https://www.neuronpedia.org/, in the appendix? We are not sure how a case study to compare the failure modes would proceed otherwise.
>
> **Question 4)** We think that this is mostly due to the fact that the human labeler remembered some examples from doing the experiment multiple times, leaking some information from other examples. We agree with the reviewer that it would be interesting to investigate this further
>
> **Question 2)** Not highlighting tokens is a very hard task both for humans and LLMs. Note that this situation is similar to detection vs fuzzing scoring of ‘Automatically interpreting millions of
> features in large language models”. It is not clear to us how attention based highlighting would be implemented. Features are either active on a specific token or they are not, and using different highlightings might lose/induce meanings that the feature does not have.
>
> **Question 3)** We have selected a single decile as the source of the 4 activating examples because this allows us to measure each decile interpretability independently. As well as do the intruder tasks with activating examples from the same feature. We have added this justification to the main text.
>
> **Question 4)** It is known that different activation functions and values of sparsity have slightly different levels of interpretability, but our method is specific for this level of sparsity. Although we did not have time to test the correlation between human scores for all the models we have added to the manuscript the correlation between the different scores for SAEs with k=64 as well as JumpRelu saes with average L0 of 56, observing that these are very similar to the ones we measure with k=32.

---

> > ### Comment · Reviewer_Bo6j · 2025-11-27
> > **Rebuttal Response**
> >
> > I thank the authors for their detailed rebuttal. I have read the rebuttal and carefully reviewed all points. Based on the author's rebuttal and the reviewer comments, I am confident in my original assessment and will maintain the score.

---

> > > ### Author Response · Authors · 2025-11-27
> > >
> > > Given that the current score of the reviewer is still that of rejection, and given that we have answered most of the reviewers questions we would like to ask if there is any pressing issue that remains that makes the reviewer not want to raise their score, specially given the scores in the soundness, presentation and contribution. Is there any weakness that the reviewer considers to remain unadressed?

---

### Official Review · Reviewer_VHFv · 2025-11-01

**Soundness:** 2
**Presentation:** 2
**Contribution:** 1
**Rating:** 2
**Confidence:** 3

**Summary:**

The authors introduce two novel methods for evaluating interpretability of SAE latents without the requirement for generative methods. The authors construct an intruder detection task as the first approach and compare performance of human and LLM detectors, showing a high correlation albeit at a small sample size of 56 latents. Example embedding scoring, on the other hand, measures proximity of positive and negative sentence samples in the latent space. Example embedding scoring reports a moderate correlation with human scores, which could be caused by the fact that sentence embedders might poorly reflect individual token relevances.

The problem of evaluating SAE interpretability is an important one, and the proposed methods have merit. The high correlation between human and LLMs on the intruder detection task is promising. The presentation of the paper, however could be improved upon. I find more extensive experiments lacking, such as adding more latents in the intruder detection task or performing subsequent analses on what causes the low correlation between human and example embedding scores — is the embedder quality a factor driving this gap? Furthermore, it is not clear to me where the data used for positive and negative SAE samples is sourced from, which is a crucial detail. An interesting question would also be also how do the methods fare across different data domains of source text?

**Strengths:**

- The authors study an interesting problem of interpreting SAE latents without the use of generative LMs
- The authors propose two methods, one of which exhibits a high correlation with human annotators

**Weaknesses:**

- The presentation of the paper would benefit from improvement
- Some important experimental details are missing: where is the data used as positive/negative samples for SAE latents sourced from?
- Experimental limitations: increasing the number of latents, or analysing the cause of poor correlation between the example embedding method and human scoring would be interesting.

**Questions:**

See above

---

> ### Author Response · Authors · 2025-11-25
>
> We want to thank the reviewer for the time they spent reviewing our work. We will reply to some points raised in the summary and address some of the weaknesses and questions.
>
> > (...) showing a high correlation albeit at a small sample size of 56 latents.
>
> We have expanded our sample size to 105 latents, which we believe to be representative of the interpretability distribution found on a normal SAE. Doubling the number of latents considered did not significantly change the correlation between the different scoring methods.
>
> > Experimental limitations: increasing the number of latents
>
> We want to highlight that our results were computed on more than 100 latents on 4 different layers, and depending on the experiments we have evaluated in total more than 1000 latents - we have updated the text to indicate the exact numbers in each experiment. Due to concerns raised by other reviewers we have also expanded our experiments to include more models: Pythia, Gemma. The correlation results are in tables A2-3 in the appendix.
>
> > Example embedding scoring reports a moderate correlation with human scores, which could be caused by the fact that sentence embedders might poorly reflect individual token relevances.
>
> Since publishing this paper, we have found some evidence that fine tuning the embedding model to perform this task on a held out set significantly improves the correlation. After fine tuning using contexts from around 300 of features of a different SAE, with a contrastive loss, we find that the correlation increases to 0.73. We want to note that the embedding model was not finetuned on any human labeled data, and with the increase of the number of human labeled datapoints, the correlation between example embedding scoring and human intruder detection is higher than that of some of the LLM correlations. The finetuning details are given in the ‘embedding finetuning’ section in the methods, having moved the old results to table A1 in the appendix.
>
> > Some important experimental details are missing: where is the data used as positive/negative samples for SAE latents sourced from?
>
> We use examples from the SmolLM2 training corpus to train the SAE. We use the same corpus to collect activations for the SAE, providing both the positive and negative samples. We clarify this in the methods section.
>
> We would love to hear suggestions from the reviewer on how to possibly improve our paper presentation so that they would be inclined to give a higher score.

---

### Author Response · Authors · 2025-12-02
**Unified Response to Reviewers**

We sincerely thank all reviewers (VHFv, Bo6j, 6pUU, Kyqc) for their time, thorough feedback, and constructive suggestions. We hope this unified response can help the AC to make a decision regarding this manuscript. In it, we have summarized the main weaknesses and questions raised and address them together, with the action taken to improve the manuscript.

**Summary of Major Weaknesses and Responses**

1. Experimental Scope and Generalizability (VHFv, Bo6j, 6pUU)

**Weakness**: Initial experiments were perceived as focusing exclusively on a small model (SmolLM2 135M) and a limited number of latents (56 human-labeled features), raising concerns about generalizability to larger models, different architectures, or alternative SAE training methods.

Changes made:
- Clarification regarding small sample size: There was a misunderstanding regarding the number of latents evaluated. The majority of our non-human-labeled experiments already used at least 100 latents per layer, totaling over 1,000 latents for some of the experiments. We have significantly updated the text to clearly state the exact number of latents used in each experiment.
- Expanded Human Trials: The number of human-labeled latents has been increased from 56 to 105. The results presented remain unchanged, and we still observe a high correlation between human and LLM scoring.
- Wider Model Evaluation: We have expanded our analysis to include SAEs trained on different models, specifically Gemma 2 9B and Pythia 160m. Our analysis now also includes different SAE activation functions and sparsity settings. The Gemma SAEs use JumpReLU with a target L0 of 51, and the Pythia SAEs have k = 64. We show that correlations remain consistent across these diverse models (Table A2-A3 in the Appendix).

2. Performance of Example Embedding Scoring (VHFv, Bo6j, 6pUU)

**Weakness:** The Example Embedding Scoring method initially showed weak correlation with human scores ($r=0.48$) and AUROC scores barely above random, undermining its utility as a scalable alternative.

Changes made:
- Clarification on AUROC: We clarified that the close-to-random AUROC is observed only in the specific case where the intruder is a low-activating example of the same feature from a different activation decile. The AUROC is higher when the intruder example is drawn from a different feature altogether.
- Improved Correlation via Fine-tuning: We have successfully addressed the weak correlation by fine-tuning the embedding model using a contrastive loss on a held-out set of SAE features. This intervention has increased the correlation with human intruder detection scores to $r=0.73$, making it a much more viable score. The details are now in the 'embedding fine-tuning' section of the methods.

3. Clarity and Motivation for Methodological Choices (VHFv, Kyqc, 6pUU)

**Weakness:** Important experimental details were missing (e.g., data source for positive/negative samples), and the motivation for key methodological choices (e.g., sampling from a single activation decile) lacked initial clarity. The introduction's motivation for avoiding natural language explanations was also deemed unclear.

Changes made:

- Data Source: We now explicitly state in the methods section that both positive and negative samples are sourced from the SmolLM2 training corpus.
- Decile Sampling Rationale: We have expanded the motivation for sampling activating examples from a single decile. This choice allows us to: 1) independently measure interpretability across the activation distribution, and 2) perform the specialized intruder task that assesses similarity between quantiles (Section 4.1.1).
- Introduction Polishing: We have refined the introduction to better articulate why avoiding natural language explanation generation is critical. This approach removes the additional hyperparameters and prompt engineering overhead required by explanation models, thereby improving the reliability of the interpretability assessment by focusing only on the SAE latent itself.


We are confident that these revisions address the concerns raised by the reviewers and significantly improve the clarity, scope, and impact of our work.

---

### Meta-Review · Area_Chair_bjN4 · 2026-01-07

**Summary:**

This work introduces two new methods for evaluating sparse autoencoders without the need to rely on explanations as done in prior work. The experimental validation supports the claims, albeit the initial set of experiments were considered too small scale and were missing details. Several reviewers pointed to these issues. Another major concern raised by the reviewers was the poor performance of one of the proposed methods, more specifically, the fact that the embedding scores do not correlate as strongly with human intruder scores.

Additional concerns raised by individual reviewers:
- Bo6j: No discussion of the failure modes.
- VHFv: Methods rely on extensive data collecting process.
- KYqC: Presentation.

**Reviewer Concerns:**

The authors addressed the following two major concerns in the rebuttal.

1/ They clarified a misunderstanding regarding the number of latents evaluated. This addressed the concern raised by several reviewers about the perceived use of only small models and a limited number of latents in the experiments. Hence, the authors' clarifications provided supporting evidence that the proposed approaches generalise to larger setting.

2/ The poor performance of the example embedding scoring method, one of the two methods introduced in this work. The authors traced back this unexpected result and proposed a fix during the rebuttal period.

The additional individual concerns were mostly addressed with the clarifications provided by the authors.

**Reviewer Scores:**

All reviewers voted initially for rejection with a relatively high confidence. However, the concerns raised by the reviewers mostly overlapped. The authors provided convincing arguments and additional details to resolve those issues. Hence, I would have expected the scores of several reviewers to increase making this a borderline accept.

---

### Decision · Program_Chairs · 2026-01-26

Accept (Poster)